# Apple Cubes Drying and Rehydration. Multiobjective Optimization of the Processes

**Radosław Winiczenko, Krzysztof Górnicki \*** , **Agnieszka Kaleta,**
**Monika Janaszek-Mańkowska** , **Aneta Choińska and Jędrzej Trajer**

Department of Fundamental Engineering, Warsaw University of Life Sciences, Nowoursynowska 164 St.,
02-787 Warsaw, Poland; radoslaw_winiczenko@sggw.pl (R.W.); agnieszka_kaleta@sggw.pl (A.K.);
monika_janaszek@sggw.pl (M.J.-M.); aneta_choinska@sggw.pl (A.C.); jedrzej_trajer@sggw.pl (J.T.)
\* Correspondence: krzysztof_gornicki@sggw.pl; Tel.: +48-22-593-46-18

**Abstract:** The effect of convective drying temperature ($T_d$), air velocity ($v$), rehydration temperature ($T_r$), and kind of rehydrating medium (pH) was studied on the following apple quality parameters: water absorption capacity (WAC), volume ratio (VR) color difference (CD). To model, simulate, and optimize parameters of the drying and rehydration processes hybrid methods artificial neural network and multiobjective genetic algorithm (MOGA) were developed. MOGA was adapted to the apple tissue, where the simultaneous minimization of CD and VR and the maximization of WAC were considered. The following parameters range were applied, $50 \leq T_d \leq 70\ °C$ and $0.01 \leq v \leq 6\ m/s$ for drying and $20 \leq T_r \leq 95\ °C$ for rehydration. Distilled water (pH = 5.45), 0.5% solution of citric acid (pH = 2.12), and apple juice (pH = 3.20) were used as rehydrating media. For determining the rehydrated apple quality parameters the mathematical formulas were developed. The following best result was found. $T_d$ = 50.1 °C, $v$ = 4.0 m/s, $T_r$ = 20.1 °C, and pH = 2.1. The values of WAC, VR, and CD were determined as 4.93, 0.44, and 0.46, respectively. Experimental verification was done, the maximum error of modeling was lower than 5.6%.

**Keywords:** optimization; genetic algorithm; artificial neural network; apple; drying; rehydration

---

## 1. Introduction

Drying is one of the most common and the oldest methods of biological materials (fruits and vegetables) preservation. It consumes between 7 and 15% of total industrial energy production [1].

Dehydration is a complex process involving moisture removal. Two processes occur simultaneously during drying, namely energy transfer (mostly of heat) from the surrounding environment to the wet solid and mass transfer (moisture transfer) from inside of the solid to the surface and then its evaporation to the surrounding environment [2].

The objective in drying of biological materials is the reduction of the amount of free-water in the solids to such a level, at which deteriorative processes caused mainly by microbiological growth, chemical reactions, and enzymatic activity are greatly minimalized. Due to the initial moisture content of approximately 74–90% w.b., vegetables and fruits are particularly susceptible to deteriorative processes [3].

During drying, however, disadvantageous changes in the material quality occur, among others: color changes due to nonenzymatic and enzymatic browning reactions, shape and size changes, shrinking, changes in texture, aromas loss, changes of the crystalline structure, hindered rehydration, lipids oxidation, protein denaturation, and loss and degradation of nutritional compounds e.g., vitamins, phenolic compounds, carotenoids, and ascorbic acid [4,5]. Therefore, the proper choice of

drying parameters, not only due to energy consumption, but above all for the quality of the final product is so important.

Rehydration is very important quality property for died products. It is a complex process intended to restore the properties of the fresh product by contacting dried products with a liquid [6,7]. The following physical mechanisms occur during the rehydration, water imbibition, internal diffusion, and convection through large open pores and at the surface. Two cross-current mass fluxes take part in the previously discussed process: a water flux from the rehydrating medium to the product and solutes flux (acids, sugars, vitamins, and minerals) from the product to the medium [8–10]. Powder–water interactions during rehydration are divided into steps: wetting, sinking, dispersing and, when the product is soluble, dissolution [11].

The process of rehydration is influenced by the following intrinsic factors, product chemical composition, its microstructure [12], predrying treatment [13,14], dehydration methods [15,16], extrinsic conditions [17,18] such as the composition of rehydrating medium [19,20], temperature [21,22], and hydrodynamic conditions [23].

Genetic algorithms (GA) are optimization methods useful in irregular experimental regions. This optimization tool is applied in such ranges as computer programming, forecasting, image recognition, control, optimization of mechanical and electronic systems, data analysis, and production management and engineering [24]. The computation efficiency of genetic algorithms could be significantly improved by their interoperating with artificial neural networks (ANN), data exploration, and fuzzy systems. ANN and GA were used for fruit storage process optimization [25]. It was found that such an intelligent approach gave better results than traditional computational techniques.

Most optimization studies not only in food industry consider to single-objective optimization, whereas multiobjective optimization (MOO), due to its mathematical complexity, has been rarely used [26,27]. An effective hybrid multiobjective evolutionary algorithm for the energy-efficient scheduling problem [28,29] and for the the passive vibration suppression of an engine [30] were proposed.

MOO is used in food industry for optimization of pre-fry microwave drying of French fries [31], convective drying of apple cubes [32], and in the process of thermal sterilization [33]. Thakur et al. [34] used MOO to balance cost and traceability in bulk grain handing, whereas Hadiyanto et al. [35] applied MOO to improve the product range of a baking system. Multiobjective optimization has been proposed for the roasting processes of beef [36] and turkey breast [37].

The aims of the study were to apply the MOO method to optimize the quality of rehydrated apples parameters by determining factors of the drying and rehydration processes. The effect of drying air temperature, drying air velocity, and the kind and temperature of the rehydrating medium on the following quality parameters of rehydrated apples was evaluated, color, volume, and water absorption capacity index.

## 2. Materials and Methods

### 2.1. Material

Apples (var. Ligol) were purchased at a Warsaw market. Fresh and high-quality lots were chosen (initial moisture content ca. 85% w.b.). Just before drying, washed and peeled apples were cut into $10 \pm 1$ mm cubes thickness.

### 2.2. Drying Process

The methods used for drying of raw apples were follows: natural convection (the drying air velocity $v = 0.01$ m/s), forced convection ($v = 0.5$ and $v = 2$ m/s), and fluidized bed drying ($v = 6$ m/s).

The drying experiments were carried at following drying air temperatures ($T_d$): 50, 60, and 70 °C. The final moisture content of dried material was ca. 9% w.b. Drying equipment and way of conducting the experiments were described in the papers [38–40].

The dried apples obtained at the same drying conditions from the three independent experiments were mixed and stored for further analysis in a sealed container for one week at the temperature 20 °C.

### 2.3. Rehydration Process

The rehydration process of dehydrated apple cubes was carried out at temperatures ($T_r$): 20, 45, 70, and 95 °C in the following media, distilled water (pH = 5.45), 0.5% solution of citric acid (pH = 2.12), and apple juice (pH = 3.20). The rehydration of samples lasted from 6 h ($T_r$ = 20 °C) to 2 h ($T_r$ = 95 °C) and was carried out in triplicate. The temperature of rehydrating liquids was constant. The initial mass of each dried sample used in rehydration amounted to ca. 10 g. Mass of dehydrated apple cubes to rehydrating medium mass ratio at the beginning of the rehydration amounted to 1:20. The values of rehydrating media pH were measured using a pH-meter, BW 10 (Trotec GmbH & Co. KG, Heinsberg, Germany) with 0.02 resolution.

### 2.4. Mass and Volume Measurements

Samples mass was measured using WPE 300 scales (RADWAG, Radom, Poland) with 0.001 g accuracy; the dry matter of solid was measured before, during, and after rehydration in accordance with AOAC standards [41]. Measurements were made in three replicates.

Volume of samples (dried and rehydrated) was calculated from buoyancy in petroleum benzine [42]. Measurements were carried out in triplicate (maximum relative error lower than 5%).

### 2.5. Color Determination

The color of the food product is one of the most important quality factors and plays a significant role in its appearance and consumer acceptability. The color of fresh and rehydrated apples was determined by scanner (Canon CanoScan 5600F). Obtained color images were then loaded into the sRGB color space. Mean brightness of pixels in each RGB channel of the image was used to express color parameters. Mean RGB values were linearly transformed to CIE XYZ color space and next XYZ color parameters were nonlinearly converted to CIE Lab coordinates. Reference values for XYZ (standard observer of 10°, illuminant D50) were 96.72, 100, and 81.43, respectively [43]. Chroma (C) and hue angle (*h*) specific for CIE*LCh* color space were determined [44].

### 2.6. Quality Parameters

The following parameters were used for description of the quality of rehydrated apple cubes,

- Water absorption capacity index (WAC) calculated from the formula [45]:

$$\text{WAC} = \frac{M_r(100 - s_r) - M_d(100 - s_d)}{M_0(100 - s_0) - M_d(100 - s_d)} \tag{1}$$

where, *M*—the mass (g), *s*—the dry matter content, and subscripts 0, *d*, and *r* refer to before drying, dry, and rehydrated, respectively. Discussed index WAC gives information on the ability of the material to absorb water.

- The volume ratio (VR) is formulated as

$$\text{VR} = \frac{V_d}{V_r} \tag{2}$$

where, $V_d$—volume of dried apple cube (after drying) and $V_r$—volume after rehydration in m$^3$.

- Color difference (CD) between the fresh and rehydrated samples determined as [46]

$$\text{CD} = \sqrt{\left(\frac{\Delta L}{K_L S_L}\right)^2 + \left(\frac{\Delta C}{K_C S_C}\right)^2 + \left(\frac{\Delta H}{K_H S_H}\right)^2} \tag{3}$$

where: $S_L$, $S_C$, and $S_H$ are the weight functions adjusting internal non-uniform structure of CIE*Lab* and $S_L = 1$, $S_C = 1 + 0.045C$, and $S_H = 1 + 0.015C$, whereas $K_L$, $K_C$, and $K_H$ (equal to 1) describe the variation from the reference conditions, $\Delta H$, $\Delta L$, and $\Delta C$ describe the difference between tested (T) and standard (S) samples in hue, luminance, and chroma, respectively. $\Delta H = 2\sqrt{C_T \cdot C_S} \cdot \sin\left(\frac{\Delta h}{2}\right)$, $\Delta L = L_T - L_S$ and $\Delta C = C_T - C_S$.

## 2.7. Quality Parameters Modeling Using ANN

Each of the ANN layers (input, output, and hidden) is built from neurons (nodes) and is fully connected to the next layer [47]. Input layer produces linear function which is a weighted sum of input variables. The hidden layer processes data with a nonlinear transfer function. The output layer processes data with a linear or nonlinear transfer function [48]. A backpropagation (BP) algorithm has been employed. The ANN has four neurons in the input layer (parameters of the drying and rehydration processes: $T_d$, $v$, $T_r$, and pH) and three neurons in the output layer (quality parameters: CD, VR, and WAC). Values of these parameters have been normalized in the range of 0–1.

In order to gain the optimal ANN architecture different number of neurons and activation functions were tried. Details of choice the best of ANN structure are described in a previous work [49]. Mean Squared Error (MSE) was calculated by

$$\text{MSE} = \frac{\sum\limits_{i=1}^{N} \left(x_{\text{exp},i} - x_{pred,i}\right)^2}{N} \tag{4}$$

where, $N$—the test cases number, $x$—the output value, subscripts exp and pred refer to experiment and prediction, respectively, and correlation coefficients (R) were used to check the performance of each ANN.

Chosen cases (114) were randomly divided into training—77 samples (70%), validation—17 samples (15%), and testing—17 samples (15%) sets. ANNs were implemented in MATLAB and the Levenberg–Marquardt algorithm was used for training [50].

To identify the critical parameters and their degree of impact on the ANN outputs, sensitivity analysis was performed. The backward stepwise method was used. This method consists of step by step rejecting one input variable, and testing the effect on the output results. The largest value of MSE due to one input omission shows the most important input [51].

## 2.8. Multiobjective Optimization (MOO) Problem

The following MOO task was taken: the determination of set of optimal conditions of drying and rehydration processes. The WAC function was maximized, whereas CD and VR functions were minimized subject to constraints on the drying and rehydration parameters. Equation (5) presents the MOO problem.

$$\text{Min}(x) = \begin{cases} \text{Max WAC}(T_d, v, T_r, pH) \\ \text{Min VR}(T_d, v, T_r, pH) \\ \text{Min CD}(T_d, v, T_r, pH) \\ 50\,^{\circ}\text{C} \leq T_d \leq 70\,^{\circ}\text{C} \\ 0.01\,m/s \leq v \leq 6\,m/s \\ 2.12 \leq pH \leq 5.45 \\ 20\,^{\circ}\text{C} \leq T_r \leq 95\,^{\circ}\text{C} \end{cases} \tag{5}$$

The Pareto front for discussed MOO problem was generated using an elitist nondominated sorting genetic algorithm (NSGA II) and was implemented in MATLAB (Figure 1).

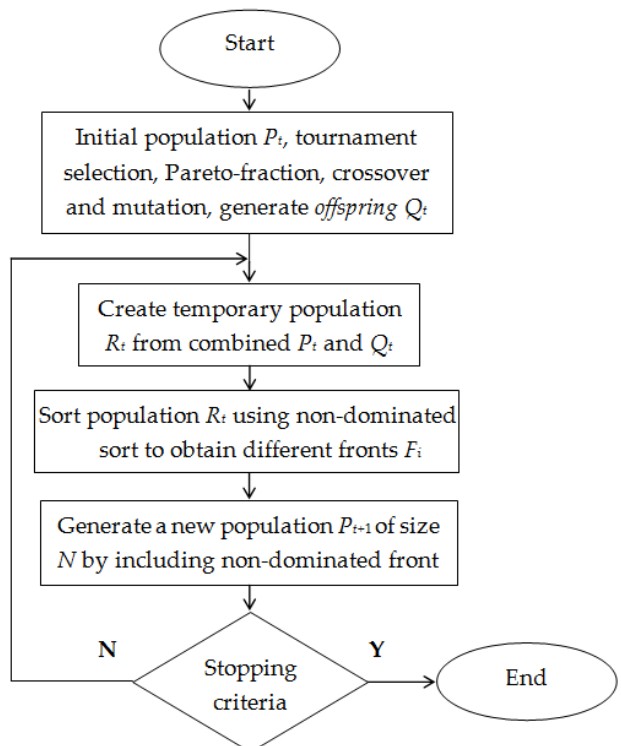

**Figure 1.** Flowchart of the elitist nondominated sorting genetic algorithm used (NSGA II).

The optimization procedure stopped at function tolerance equal to $10^{-6}$.

## 3. Results and Discussion

### 3.1. ANN

In order to approximate functional relations between drying and rehydration processes variables ($T_d$, $v$, pH, and $T_r$) and apple quality parameters (WAC, VR, and CD), different ANN structures with various transfer functions were tested. Considering the highest R and the lowest MSE the best result (MSE = 0.0019) were obtained for ANN presented in Figure 2. The hidden and output layers of the ANN processed data with a log-sigmoid transfer functions.

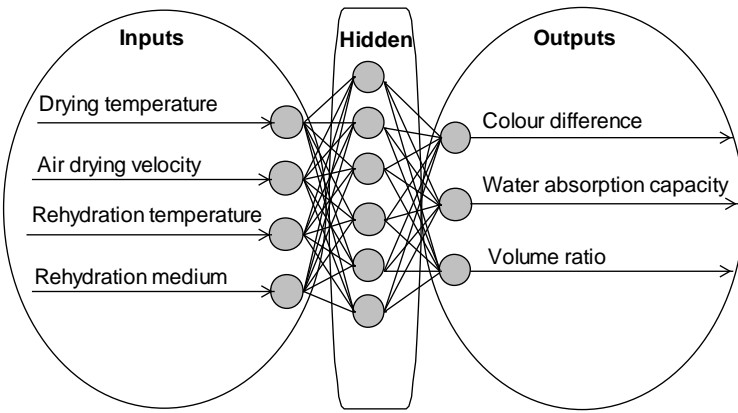

**Figure 2.** The best of artificial neural network (ANN) structure.

Figure 3 shows the ANN best validation performance. MSE = 0.0019 at the 16th iteration with changes of MSE at training, validation, and testing phase. Mean squared error determined for test and validation sets had similar characteristics. Insignificant overfitting was observed. Final MSE values for training, validation and test sets were 0.0014, 0.0019, and 0.0017, respectively.

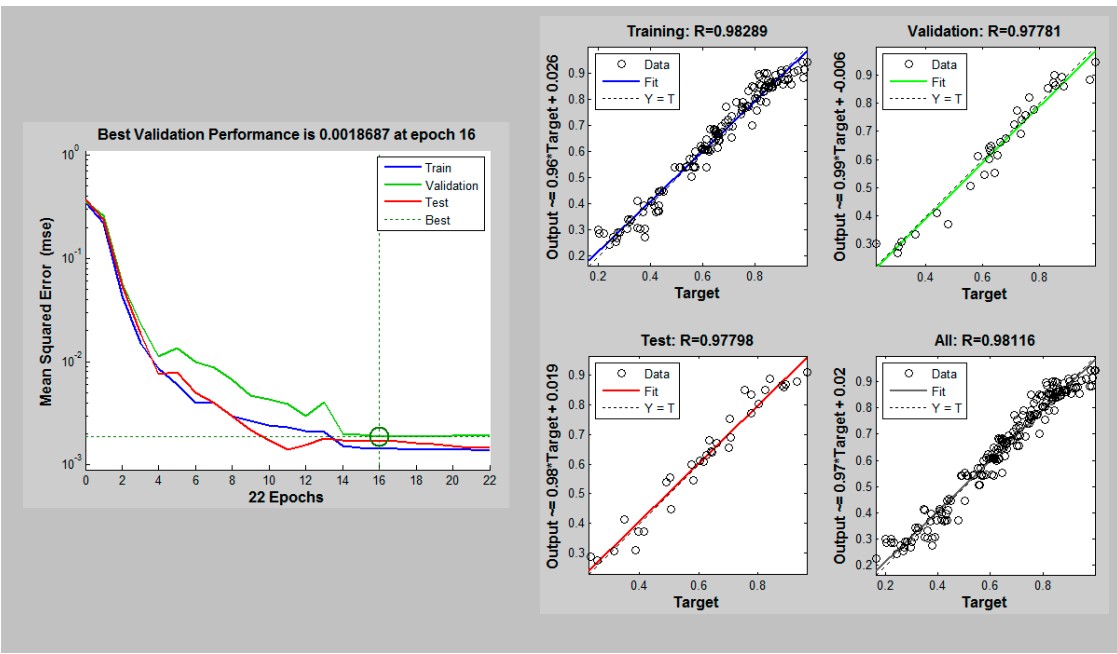

**Figure 3.** ANN best validation performance (**a**) and ANN goodness of fit (**b**).

Additionally, high R-values between predicted and experimental data (0.9778–0.9829) mean good agreement between data predicted by ANN and experimental results (Figure 3).

### 3.2. Mathematical Formulations

Quality parameters WAC, VR, and CD were determined with the following formulas (from ANN).

$$\text{WAC} = \frac{1}{1 + \exp^{-(1.9043F_1 - 1.8376F_2 + 2.0638F_3 + 0.6350*F_4 - 1.0502F_5 + 1.3317F_6 - 2.1991)}} \tag{6}$$

$$\text{VR} = \frac{1}{1 + \exp^{-(-0.7714F_1 + 0.8390F_2 + 4.3547F_3 - 2.2097F_4 - 2.0998F_5 - 0.4509F_6 - 1.0072)}} \tag{7}$$

$$\text{CD} = \frac{1}{1 + \exp^{-(-3.5956F_1 + 0.6489F_2 - 0.3165F_3 - 3.0398F_4 + 4.5572F_5 - 1.1678F_6 + 5.8421)}} \tag{8}$$

where $F_{(i=1 \div 6)}$

$$F_i = \frac{1}{1 + \exp^{-W_i}} \tag{9}$$

and $W_1 - W_5$ can be determined as follows

$$W_i = D_{1i}T_d + D_{2i}v + D_{3i}\text{pH} + D_{4i}T_r + D_{5i} \tag{10}$$

where constants $D_{ji}$ in Equation (10) are shown in Table 1.

**Table 1.** Weights and biases in Equation (10).

| $i$ | $D_{1i}$ | $D_{2i}$ | $D_{3i}$ | $D_{4i}$ | $D_{5i}$ |
|---|---|---|---|---|---|
| 1 | −11.1588 | −7.7592 | −0.6697 | 1.5380 | 18.4940 |
| 2 | −1.0835 | −0.1818 | −9.6281 | −3.8191 | 10.5589 |
| 3 | −15.0529 | −4.9146 | 8.6942 | −2.8797 | 17.9134 |
| 4 | −8.7055 | −0.6294 | 4.3391 | −14.9923 | 17.1001 |
| 5 | −11.8990 | −18.2054 | −1.8929 | 2.3569 | 9.2392 |
| 6 | −3.6760 | 1.6062 | −17.8052 | −5.5386 | 11.8517 |

The normalization of the $T_d$, $v$, pH, and $T_r$ values were conducted dividing them by 70, 6, 5.5, and 95, respectively, whereas the values of quality parameters (WAC, VR, and CD) were normalized by dividing them by 0.54, 0.64, and 27 respectively. Equations (6)–(10) were used for MOO (Equation (5)).

The sensitivity analysis showed that $T_d$ has the greatest impact on all the quality parameters obtained from the ANN. $V$, pH, and $T_r$ occupied the 2nd, 3rd, and 4th position, respectively. In the case of testing the sensitivity analysis of ANN only for CD, $T_d$ has the greatest impact on this quality parameter and next positions occupied the $T_r$, $v$, and pH. However, when the ANN is used to designate only VR, the greatest impact on VR is in the following sequence: pH, $T_r$, $v$, and $T_d$. Taking into account only the WAC, the order is as follows pH, $v$, $T_r$, and $T_d$.

## 3.3. MOO

The MOO problem (Equation (5)) was solved with GA. The size of population was assumed as 30. The controlled parameters of NSGA II were as follows. The mutation function was Adaptive feasible and the crossover function was Heuristic with default ratio of 1.2. Number of generations was 300 and Pareto front population fraction was 0.8.

Table 2 and Figure 4 show 24 design points of the Pareto set and the nondominated points of Pareto front, respectively.

**Table 2.** Pareto optimal set given in random order.

| Pareto ID | WAC (−) | VR (−) | CD (−) | $T_d$ (°C) | $v$ (m/s) | pH (−) | $T_r$ (°C) |
|---|---|---|---|---|---|---|---|
| 1 * | 0.4610 | 0.4406 | 4.9339 | 50.0726 | 4.0269 | 2.1231 | 20.0787 |
| 2 | 0.2496 | 0.5829 | 26.3198 | 59.1482 | 0.3961 | 2.1268 | 84.6113 |
| 3 | 0.3505 | 0.5290 | 14.8788 | 56.8466 | 1.4085 | 2.5738 | 67.7296 |
| 4 | 0.2987 | 0.6130 | 24.6705 | 60.4821 | 2.1437 | 2.3132 | 91.4555 |
| 5 | 0.3560 | 0.5128 | 13.2246 | 56.9539 | 1.3443 | 2.4292 | 63.7404 |
| 6 | 0.3490 | 0.5876 | 21.8093 | 57.4686 | 1.5863 | 2.9835 | 81.2848 |
| 7 | 0.4427 | 0.2771 | 20.9616 | 53.2569 | 0.1899 | 4.6265 | 72.0360 |
| 8 | 0.4767 | 0.3696 | 12.1261 | 52.1816 | 0.7663 | 4.9951 | 75.2174 |
| 9 | 0.3854 | 0.4775 | 8.8345 | 53.1772 | 3.4738 | 2.8495 | 33.9584 |
| 10 | 0.4395 | 0.3129 | 19.4062 | 53.5391 | 0.2951 | 4.4393 | 71.1104 |
| 11 | 0.4607 | 0.2376 | 20.2743 | 50.6788 | 0.1718 | 5.4014 | 66.2427 |
| 12 | 0.4494 | 0.1935 | 23.0974 | 50.0522 | 0.0151 | 5.3859 | 66.7517 |
| 13 | 0.4884 | 0.3929 | 7.2222 | 61.1844 | 1.9569 | 5.4500 | 59.5787 |
| 14 | 0.4310 | 0.2173 | 23.7054 | 52.1683 | 0.0145 | 4.7001 | 71.9461 |
| 15* | 0.4459 | 0.4501 | 5.7222 | 51.7956 | 3.3349 | 2.2060 | 31.0002 |
| 16 | 0.4519 | 0.3629 | 16.3099 | 55.2706 | 0.4085 | 4.4603 | 72.3659 |
| 17 | 0.3722 | 0.4924 | 10.4412 | 55.9213 | 1.7468 | 2.3832 | 57.1886 |
| 18 | 0.3659 | 0.4950 | 10.9728 | 56.3231 | 1.6107 | 2.4896 | 58.2014 |
| 19 | 0.4725 | 0.4049 | 8.1156 | 55.7922 | 1.1456 | 4.8176 | 57.8621 |
| 20 | 0.4665 | 0.3310 | 14.7560 | 53.9473 | 0.4089 | 4.8891 | 69.1178 |
| 21 | 0.3001 | 0.6125 | 24.6999 | 60.7488 | 2.4932 | 2.3504 | 92.8243 |
| 22 | 0.2810 | 0.5220 | 26.1803 | 57.1793 | 0.2308 | 2.8994 | 81.7101 |
| 23 | 0.2053 | 0.5559 | 26.8543 | 60.0454 | 0.0122 | 2.1271 | 91.4352 |
| 24 | 0.4432 | 0.3274 | 18.2983 | 53.5391 | 0.3576 | 4.4393 | 71.1104 |

* Best solution.

Conflicting relations between all the objective functions of drying and rehydration processes quality have been observed, therefore finding a solution that simultaneously optimizes all taken quality parameters is not possible. Figure 4 shows that the increase in CD causes both increase and decrease in VR (clearly larger). The WAC index increases slightly or decreases (especially for large CD) as the CD increases. The WAC index increases slightly, then drops sharply with the increase in VR.

As far as the CD is concerned the best solution is for ID 1 (CD = 4.93), however for WAC and VR the best solution can be assumed at ID 13 (WAC = 0.49) and ID 4 (VR = 0.61), respectively.

In case of the smallest value of CD (ID 1) where VR = 0.44 and WAC = 0.46, the optimum values of drying and rehydration processes variables were $T_d$ = 50.1 °C, $v$ = 4.0 m/s, rehydrating medium: solution of citric acid (pH = 2.12), rehydrating temperature 20.1 °C. At these conditions the value of VR was 28.1% smaller than the greatest VR (ID 4) and the value of WAC was 5.6% smaller than the greatest WAC (ID 13). A slightly worse result of optimal solution was obtained for ID 15. The values of WAC, VR, and CD were 0.45, 0.46, and 5.7, respectively. It can be stated that at ID 15 the value of CD was 16.0% greater than the smallest CD (ID 1), VR was 26.6% smaller than the greatest VR (ID 4), and WAC was 8.7% smaller than the greatest value of WAC (ID 13).

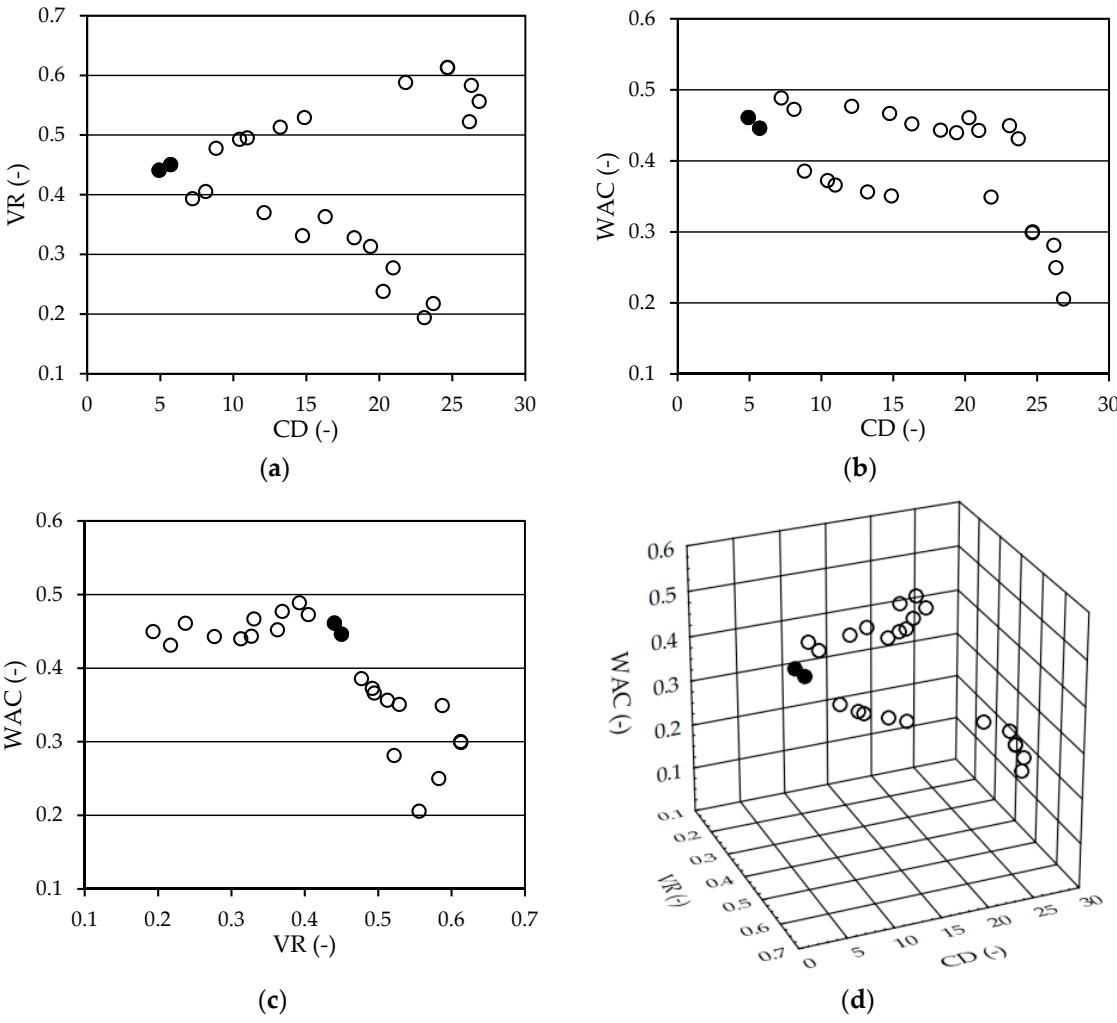

**Figure 4.** Two-dimensional (**a**–**c**) and three-dimensional (**d**) views of Pareto front; (●)—the best solutions.

However, taking into account the greatest value of VR (ID 4) for which CD = 24.67 and WAC = 0.30 the optimum values of the process variables were $T_d$ = 60.5 °C, $v$ = 2.1 m/s, pH = 2.21, and $T_r$ = 31.0 °C. It can be noticed that at these conditions the value of CD was 400% greater than the smallest CD (ID 1) and WAC was 38.9% smaller than the greatest value of WAC (ID 13).

Again, taking into account the greatest value of WAC (ID 13, CD = 7.22, VR = 0.39) the optimum values of the process variables were $T_d$ = 61.2 °C, $v$ = 2.0 m/s, pH = 5.45, $T_r$ = 59.6 °C, and CD was 46.4% greater than the smallest CD (ID 1), VR was 35.9% smaller than the greatest value of VR (ID 4).

It can be seen from Table 2 that point for which all taken functions (maximum WAC and VR, minimum CD) achieved simultaneously their optimum values cannot be find. Then the solution is formulated by set of nondominated solutions in Pareto sense. It can be stated therefore that the choice of one solution depends on the individual requirements.

Taking into account the differences in the obtained values for various strategies (ID 1, ID 4, and ID 13), we recommend Pareto solution as ID 1 (minimum CD).

Results of the model validation follow. The validation was done using the same values of process parameters ($T_d$ = 50.1 °C, $v$ = 4.0 m/s, $T_r$ = 20.1 °C, and pH = 2.12) to demonstrate the reliability of predicted quality parameters values. The dehydration process was carried out in a fluidized bed dryer, whereas the rehydration process was carried out in a solution of citric acid. The values of WAC, VR, and CD were 0.45, 0.45, and 5.21, respectively, and they were very close to the values predicted by ANN (0.46, 0.44, and 4.93). The maximum error of modeling was lower than 5.60%.

In our work and in the literature [32,52] various processes parameters (drying and rehydration) and various quality criteria were considered. Winiczenko et al. [52] carried out optimization of the drying and rehydration (in distilled water) processes of apple using MOGA algorithm. They obtained the following recommended processes parameters. $T_d$ = 50.1 °C, $v$ = 0.03 m/s, and $T_r$ = 67.5 °C. Moreover, Winiczenko et al. [32] conducted optimization of the drying process of apple using a NSGA II algorithm. It turned out that the following drying process parameters can be recommended, $T_d$ = 65 °C, $v$ = 1.0 m/s.

It should be stressed, however, that it turned out from the present work and from the literature that finding the conditions of the drying and rehydration processes that simultaneously optimized different quality parameters of rehydrated apple seems to be impossible. Therefore, as far as the optimization of the drying and rehydration processes is concerned, it should be determined which quality parameter of rehydrated apple is considered most important or decisive in the given situation.

## 4. Conclusions

The drying process brings about the undesirable changes in the quality of dehydrated product. The application of convenient food requires its hydration. It is important to search for methods of drying and rehydration processes optimizing to ensure good quality of dried food.

The paper used a novel MOO method (based on ANN, GA, and Pareto optimization) for optimizing of processes of apple drying and rehydration. A novel MOO GA method with consideration of the simultaneous maximization of WAC and minimization of CD and VR, as rehydrated apple quality parameters, was successfully applied.

The back-propagation algorithm for ANN training was sufficient to predict the rehydrated apple quality. The mathematical formulas (from the ANN) for determining WAC, VR, and CD were obtained. It was found that relationships between drying and rehydration processes variables and quality characteristics of rehydrated apple are nonlinear. The ANN with sigmoidal transfer function may be used for predicting the quality of rehydrated apple.

The optimum values of processes variables, gained by the MOO GA, were $T_d$ = 50.1 °C, $v$ = 4.0 m/s, pH 2.1, and $T_r$ = 20.1 °C. WAC, VR, and CD for dehydrated and next rehydrated apple at these terms: 0.46, 0.44, and 4.9, respectively. Experimental verification gave the value of the maximum error of modeling lower than 5.6%. The investigations proved that finding the conditions of the considered processes that simultaneously optimize three discussed quality parameters (WAC, VR, and CD) of rehydrated apples is impossible.

**Author Contributions:** R.W.: Formal analysis, methodology, and software, K.G.: Conceptualization, formal analysis, investigation, methodology, software, and writing of the manuscript, A.K.: Conceptualization, project administration, supervision, and writing of the manuscript, M.J.-M.: Supervision and investigation, A.C.: Investigation, J.T.: Critical review of manuscript.

**Funding:** Polish National Science Centre, N N 313 780940.

**Conflicts of Interest:** The authors declare no conflict of interest.

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
