# Peer review of "Apple Cubes Drying and Rehydration. Multiobjective Optimization of the Processes"

_sustainability, doi:10.3390/su10114126_

Round 1
Reviewer 1 Report
Comments and Suggestions for Authors
COMMENTS ON “Apple cubes drying and rehydration. Multi-objective optimization of the processes”
Content comments:
1. The manuscript is concerned with the modelling, simulation, and optimisation parameters of the drying and rehydration processes of the apple cubes; this topic is interesting. It is relevant and within the scope of the journal.
2. However, the manuscript, its present form, contains several weaknesses. The adequate revisions concerning the following points should be undertaken to justify the recommendation for publication.
3. The manuscript deals with the problem of the approximation of the functional relations between drying and rehydration processes variables and apple quality parameters by ANN. What are the advantages of applying this particular one over classical approaches? How will this affect the results? More details should be furnished.
4. How normalization values were chosen? How will this affect the results? Additional analysis should be furnished.
5. The experimental verification of the proposed method must be performed in a more detailed way.
6. Sensitivity analysis of the proposed method must be performed to detect the range and resolution of this approach. Additional analysis should be furnished.
7. The discussion section in the present form is relatively weak and should be strengthened with more details and justifications.
Author Response
Response to Reviewer 1 Comments
1. The manuscript is concerned with the modelling, simulation, and optimisation parameters of the drying and rehydration processes of the apple cubes; this topic is interesting. It is relevant and within the scope of the journal.
2. However, the manuscript, its present form, contains several weaknesses. The adequate revisions concerning the following points should be undertaken to justify the recommendation for publication.
3. The manuscript deals with the problem of the approximation of the functional relations between drying and rehydration processes variables and apple quality parameters by ANN. What are the advantages of applying this particular one over classical approaches? How will this affect the results? More details should be furnished.
The mathematical models for describing the effect of drying and rehydrating parameters and rehydratatcji on the quality parameters of apples are needed for optimization. In the available literature the description of similar relationships using simple functions was not found. Successfully employed ANN (approximating its properties) enable to describe needed relationship. The sensitivity analysis of ANN was also performed.
4. How normalization values were chosen? How will this affect the results? Additional analysis should be furnished.
Values of the parameters have been normalized in the range of 0-1. The normalization of the Td, v, pH and Tr values were conducted dividing them by 70, 6, 5.5 and 95 (the max parameter values), respectively, whereas the values of quality parameters (WAC, VR, CD) were normalized by dividing them by 0.54, 0.64 and 27 (the max parameter values), respectively. Log-sigmoid transfer returns value from 0 to 1. Lines 327-329.
5. The experimental verification of the proposed method must be performed in a more detailed way.
More details about experimental verification is now in the text - lines: 514-515.
6. Sensitivity analysis of the proposed method must be performed to detect the range and resolution of this approach. Additional analysis should be furnished.
Sensitivity analysis (backward stepwise method) was done – lines 150-153, 330-335.
7. The discussion section in the present form is relatively weak and should be strengthened with more details and justifications.
Sensitivity analysis was added to the text (lines 330-335). More details about the experimental verification included in the text (lines: 514-515). The results of the optimization of the drying and rehydration processes of apple using MOGA alghorithm and optimization of the drying process of apple using NSGA II alghorithm from the literature were added to the text. General summary form the investigation form present work and the literature is also added (lines 515-525).

Reviewer 2 Report
This paper by Winiczenko et al. optimizes the apple cubes drying and rehydration using genetic algorithm and artificial neural networks (ANN). The authors had a carefully modeling and optimization process. This is an interesting addition for the machine learning community. The research topic well-suits for the journal. The manuscript is well-prepared.
Therefore, I recommend that this paper is acceptable to the journal after a minor revision. Details of my comments are shown as follows.
Comments:
1. What was the activation function used for this study?
2. A schematic picture of the used ANN and their independent variables should be clearly shown in the paper.
3. How did the authors define the best network structure? More information should be clearly shown in the paper.
4. Is it a reason that not to use the response surface methodology for the optimization in this case?
5. There are several typos in the manuscript. Please double-check the paper.
Author Response
Response to Reviewer 2 Comments
Comments:
1. What was the activation function used for this study?
Log-sigmoid function: (line 170), eqs. (6)-(9)
2. A schematic picture of the used ANN and their independent variables should be clearly shown in the paper.
Figure 2 is now corrected
3. How did the authors define the best network structure? More information should be clearly shown in the paper.
Lines 142-146.
4. Is it a reason that not to use the response surface methodology for the optimization in this case?
The great advantage of the GA technique over the RSM is that the GAs do not need to generate models. Forbidden or unreachable combinations of the factor settings can be simply put aside with another run of the program (or, in a lees recommended way, by assuming undesirable values for the responses in that particular condition). Therefore, it is likely that the GA technique provides better results, over the RSM technique, in an irregular experimental region.
5. There are several typos in the manuscript. Please double-check the paper.
Paper was checked

Round 2
Reviewer 1 Report
The manuscript can be published.